# Strategies to Prevent and Cope with Adolescent Dating Violence: A Qualitative Study

**DOI:** 10.3390/ijerph20032355

**Published:** 2023-01-28

**Authors:** María del Mar Pastor-Bravo, Elka Vargas, Venus Medina-Maldonado

**Affiliations:** 1Faculty of Nursing, Cartagena School of Nursing, University of Murcia, 30203 Cartagena, Spain; 2Research Group of Gender Violence Prevention (E-previo), Nursing Faculty, Pontifical Catholic University of Ecuador, Quito 170143, Ecuador; 3Faculty of Psychology, Pontifical Catholic University of Ecuador, Quito 170143, Ecuador; 4Centro de Investigación para la Salud en América Latina (CISeAL), Nursing Faculty, Pontifical Catholic University of Ecuador, Nayón 170530, Ecuador

**Keywords:** primary prevention, intimate partner violence, adolescent, adaptation psychological, health education

## Abstract

Background: Dating violence has attracted scholarly interest from many fields because of its implications for adolescents’ health. This study aimed to learn which strategies adolescents use to address and prevent dating violence. Methods: Eight focus group discussions were analyzed, which included a total of 78 adolescents between 13 and 17 years old who had the signed consent of their parents or guardians. Results: The findings showed that the majority of adolescents lacked effective mechanisms to manage violence in their dating relationships, which were characterized by immaturity, a lack of trust in their families, and unrecognized relationship violence. Conclusions: Assessing roles and relationships, as well as coping mechanisms, is a valid way to approach adolescents and explore how they perceive interaction during dating and which strategies are used most frequently to prevent violence.

## 1. Introduction

Romantic relationships prior to marriage have existed for centuries; however, during the 20th century, love and sexuality have come to occupy a more prominent place in both research and society. As such, courtship and seduction in dating cannot be separated from cultural, economic, communicational, and contextual factors, such as philosophical, political, and birth aspects. Dating, meaning a relationship between two people, is considered the archetype of romantic relationships, although other forms of attachment exist that have led to the study of new relational dynamics that have influenced the transformation of emotional ties [1]. Dating is a dyadic relationship that involves the social interaction between two people who have the intention of sharing common experiences until one of the two individuals decides to end it or to establish another type of link, such as cohabitation or marriage [1].

Dating violence is considered a type of intimate partner violence; it can take place in person, online, or through technology, and the main forms of abusive behaviors are physical violence, sexual violence, psychological aggression, and stalking [2]. This problem has gained scholarly interest from many fields, as the frequent incidence of abusive behaviors in romantic relationships, both for teens and young adults, represents a social problem that has implications for public health. The theoretical orientation of this study draws attention from four interrelated insights, including empirical knowledge of teen dating violence, gender perspective, strategies used by adolescents to prevent dating violence, and coping theory.

Different studies have highlighted the high level of perpetration and/or victimization in relationships at these ages [3,4,5]. In terms of the form this takes, bidirectional violence is more frequent for psychological violence, but when behaviors escalate to physical violence, it becomes unidirectional [6]. A study on the most severe types of violence found that it is more often perpetrated by boys, especially girls are more likely to be victimized in cases of sexual violence [7].

Consideration of gender reproduction in society and the negative effects for women when the orientation is a patriarchal system is not new. Particularly, adolescents exposed to societal norms based on sexist beliefs tend to disproportionately idealize romantic relationships. They are prone to have misconceptions about love and normalize inequity behaviors. Current scientific literature on dating violence revealed that both girls and boys held attitudes linked to dominant power in dating relationships because of values and role expectations [8,9].

Regarding strategies to prevent dating violence, a systematic review of 10 studies of help-seeking intentions and behaviors in youths between the ages of 12 and 19 based on racial and ethnic differences found that, in most of these studies, youths relied on informal sources of support, with youths from both groups preferring to seek help from parents and friends. Mistrust, lack of closeness, shame, and embarrassment informed youths’ help-seeking intentions and behaviors. Racially and ethnically specific factors, such as negative perceptions of father figures, familism, acculturation, and traditional gender role notions, were identified as barriers to help seeking [10].

According to the literature, the two forms of coping are problem-focused and emotion-focused coping strategies [11,12]. A study in Ghana, for example, showed that in 405 adolescents with mental health difficulties, “the coping strategies used by them in dealing with the problems were isolation, use of illicit drugs and seeking spiritual help” [13]. Regarding the relevance of coping mechanisms in teen dating violence research, few studies have applied this perspective to analyze the barriers to violence prevention, particularly the psychological and emotional responses that hinder seeking help and obtaining information on healthy relationships and that normalize violence, guilt, denial, fear, and anger. 

Our study’s findings will be useful to design a future intervention proposal that promotes adolescents’ healthy behaviors related to the constructive use of coping mechanisms, as well as the promotion of egalitarian, peaceful, and respectful relationships. One way to respond effectively to this group is by exploring their perceptions about dating violence through meeting for discussions, which can be achieved through a qualitative approach. The focus group technique provides benefits in the collective subjectivity assessment and involving adolescents in identifying realistic expectations for themselves in terms of prevention. The resulting data can lead to the dissemination of clear coping strategies that reflect the interests of this age group, and the generated dialogue can stimulate reflection and identification of barriers that prevent the achievement of the desired healthy behaviors [14,15].

In this sense, this qualitative study aimed to learn which strategies are used to address and prevent dating violence from the perspective of adolescents in Quito. 

## 2. Materials and Methods

A qualitative design was used, consisting of focus groups with adolescents from different educational institutions in the city of Quito, Ecuador.

The participant selection criteria were as follows: aged between 13 and 17 years; currently have or have had a committed romantic relationship; be enrolled in one of the participating educational institutions; and, in the case of minors, have informed consent from parents or legal guardians. The exclusion criteria were cohabitation with partner, being married, and not having had any previous romantic relationship.

Convenience sampling was used to select the participants, and participation was voluntary. The participants were informed that they could withdraw from the study at any time without incurring any penalty.

The study sample comprised 78 adolescents between 13 and 17 years old from eight educational institutions in Quito. Informed consent forms were given to the parents and/or representatives of the potential participants. Participants who did not submit this form were excluded from the study.

Before beginning each focus group session, the motive and objective of the study were presented to the participants, and its voluntary nature and the confidentiality of the collected information were discussed. Consent to record the sessions was also requested. This study is part of the research project “Violence during dating: A mixed study with adolescents from Quito, Ecuador,” which received funding and approval from the Ethics Committee for Research on Human Subjects from the Pontifical Catholic University of Ecuador (no. 2018-52-EO), the National Office of Educational Research from the Ministry of Education of Ecuador (memorandum no. MINEDUC-DNIE-2019-00067-M), and the rectors of the selected educational institutions, who were informed of the project objectives.

The data were collected through mixed-gender focus groups, which included a minimum of six participants and a maximum of fourteen, to gather information on the opinions and experiences of the adolescents regarding strategies for coping with and preventing dating violence. A moderator led the discussions using a semi-structured questionnaire, and a non-participating observer conducted the interview and took notes on the interactions and nonverbal aspects. Both the moderator and the observer were researchers at the Pontifical Catholic University of Ecuador with a PhD and a Master’s degree, as well experience in the subject of gender-based violence. The focus groups were conducted from November 2019 to March 2020. The discussions were carried out using a script with semi-structured questions that guided the focus group discussions through the following main topics: (1) coping strategies for gender-based violence and (2) prevention of gender-based violence in dating relationships. The average discussion time in each of the focus groups was 60 min. During the group interviews, all participants sat in a circle to increase interaction with each other. All the interviews were audio recorded by the moderator and the observer, which were later transcribed for analysis.

After transcribing the focus group audio recordings, they were analyzed using qualitative content analysis [16]. This involved creating a categorical framework in which predetermined codes were described and defined before analysis based on the theoretical framework of this study. From the authors’ perspective “this qualitative step of analysis consists in a methodological controlled assignment of the category to a passage of text” [17] obtained in the focus groups (See Table 1).

Subsequently, the inductive categories that emerged through the modification and expansion of previously established categories and subcategories were incorporated until the final qualitative analysis was reached. Atlas.ti version 8.4 (license available through the University of Murcia) was used for the organization and systematization of the data and the construction of structural networks.

Theoretical saturation was reached in situ after using theoretical sampling to validate that no new topics arose in each phase of the discussion. The analytical process was carried out by two researchers simultaneously and independently; subsequently, the results were discussed with a third researcher until a consensus was reached through triangulation. The protocolization of data collection and the detailed description of the study sample and context enable the study methodology to be reproduced under similar conditions.

## 3. Results

### 3.1. Participant Characteristics

The data analyzed in this study correspond to eight focus groups with adolescents from Quito, Ecuador, aged 13 to 17 years old, whose school level ranged from the ninth course of basic general education (BGE) to the third year of high school. A total of 78 adolescents participated: 45 girls and 33 boys (Table 2). The composition differences among the focus groups and sex were due to the fact that several pre-selected participants came to the focus group session without a signed consent form, which meant they were excluded from the study.

Because of the desire to include a wide range of perspectives among the participants, the selected schools were diverse in terms of being public or private, being religious or secular, and being from different city locations.

The participants were identified using the letter “P,” the participant’s code number, and the initial “F” for female and “M” for male, followed by the code for their corresponding educational institution (i.e., the letter “D” for “discussion” and a number).

Regarding the dynamics of participation and interaction, the adolescents who were more willing to talk and who expressed their ideas more easily participated more. Some of the focus groups comprising adolescents (boys and girls) from lower grade levels (especially the ninth course of BGE) were less productive compared to the others, which might be related to the participants’ lack of experience and knowledge on the subject.

### 3.2. Prevention and Coping Strategies for Teen Dating Violence

Three major themes were uncovered: (1) factors that hinder the prevention of and coping with dating violence, (2) coping with dating violence, and (3) strategies for preventing dating violence.

#### 3.2.1. Factors That Hinder the Prevention of and Coping with Dating Violence

This theme included the cultural norms that explain the occurrence of intimate partner violence and how it constitutes obstacles to having a healthy relationship. The adolescents reported being in a stage of greater vulnerability to dating violence due to their immaturity; they referred to the fact that their inexperience can make it harder for them to identify dating violence when it occurs. They also said that sometimes violence begins as a game between the couple that leads to increasingly violent episodes, which may influence their understanding of relationships in the future.


*“This isn’t something that happens overnight; the victim has to realize that the violence is progressing…. At first, the behavior is hidden but later becomes more serious until it sometimes turns into physical violence” (P5M-D1).*



*“I think if the relationship continues like this, the violence can go further than physical or psychological violence…it can lead to death or suicide” (P3M-D5).*


Even physical hitting is difficult for teenagers to identify as intimate partner violence because it commonly starts as a game.


*“Because if they’re already like that as boyfriends, if they get married, then they can end up killing me or killing themselves. Because if they’re already going through this as kids, as adults, they’re going to think that this is okay and that this is how someone should or shouldn’t be treated in a relationship” (P4F-D7).*


Participants said that jealousy is one of the main games in relationships, which could lead to violence. Throughout the discussions, jealousy was characterized both as a game and a form of fun for the couple and as a demonstration of true love.


*“At first, it’s like a game, you can say: ‘Hey! Why are you cheating on me?’ Like a joke, but then over time, it can become more serious, because the two people start to attack each other” (P3F-D3).*



*“If he didn’t show any interest in me, then I do the same to him, and I’ll be with someone else, but I’m still in the relationship. So then, if he gets jealous, I know he still loves me” (P2F-D7).*


In this theme, we found some experiences related to the myths of romantic love; fear; threats to harm the victim or their family if they leave the relationship; lack of early identification of being in a violent relationship; shame and concealment of violence; and a lack of information and resources for managing already-occurring violence. Other factors typical of a patriarchal culture, such as the normalization of violence, the tendency to blame the victim, the myths of romantic love that reinforce the idea that love conquers all, the linking of love with abuse, the justification of violence, and the repetition of sexist patterns learned at home, make it difficult to take action against dating violence. The participants also indicated that their own inexperience, lack of communicational resources to resolve conflict, and lack of trust and communication with family can be obstacles to asking for help. The testimonials have been classified into categories and subcategories (see Table 3). 

#### 3.2.2. Coping with Dating Violence

This theme was related to behavioral tactics used by the adolescents to manage violence in a dating relationship. The adolescents shared their experiences, and they identified some important aspects such as avoiding isolation, communicating the situation, asking for advice, turning to mothers or people with more experience, and reporting the situation to professionals if it is happening to someone else (Table 4).

Some adolescents described their experiences with asking for help to leave a violent dating relationship. Asking for help is the most important skill when adolescents are dealing with a difficult interpersonal relationship because it increases the awareness about violence, courage, and trust building.

However, other adolescents used strategies based on trying to make it better and continuing with the relationship. They identified one way was changing their own behavior to please their partner and, thus, encourage them to abandon the violent attitudes in order to stay in the relationship. Another strategy involved asking for forgiveness to avoid starting an argument and encouraging communication within the relationship.


*“I would try to change to see if he wanted to change. So, the relationship could continue” (P3F-D6).*



*“What I’ve seen in relationships is pride. Quite a few break up, arguing or fighting over pride, let’s say that if one person is to blame, s/he should be able to apologize for causing for the problem” (P4M-D3).*



*“Talk to your partner and if, like, if he doesn’t want to understand, break up with him” (P2F-D6).*


#### 3.2.3. Strategies for Preventing Dating Violence

This theme described actions that the adolescents identified as necessary to prevent violent dating relationships, and these can be considered personal skills, such as having fluid communication with the family. They also identified the importance of talking about their dating experience with more experienced individuals who could give them guidance on the relationship, including obtaining examples that are passed on at home, education, and specific information they can transmit to others from family members, regarding healthy relationships and gender-based violence. Additionally, they recognized the importance of creating spaces for dialogue about intimate partner violence; receiving related training for both themselves and their families; and not falling to social pressure to remain in a violent relationship, which breaks with the mandate of the patriarchal system (Table 5).

The adolescents emphasized that an educational intervention about this topic must be developed with attractive methodologies to grab the attention of their age group. They recommended that training, workshops, discussions, dramatizations, and talking about the experience rather than theorizing about the problem are especially important for making the discussions more dynamic; these also facilitate people’s identification with different situations that can lead to violence, so these can be recognized when such situations are experienced (Table 6).

The adolescents perceived that to have a healthy relationship, the following factors are necessary: dialogue and fluid communication, trust in the partner, mutual respect, a balance between what you give and what you receive, setting boundaries, taking time to get to know each other, respecting the partner’s hobbies and space, and being honest with each other. Similarly, some participants emphasized the importance of respecting and loving oneself before and during the relationship. They also said that through the promotion and knowledge of what a healthy dating relationship entails, violence can be prevented.

## 4. Discussion

One of our remarkable findings involved coping mechanisms. Specifically, we found that most of the adolescents had ineffective ways to manage violence in dating. In our study, some adolescents who had experienced threats of violence, fear, shame, and concealment of violence reported a lack of knowledge of how to act appropriately. Besides, adolescents’ lack of assertive communication skills might decrease their ability to identify conflicts, seek solutions, and obtain support from family or a significant adult when these conflicts exceed their coping abilities. These factors may be contributing to the lack of help-seeking behavior among victims of teen dating violence detected in another article [18], but they also are affected by the lack of special programs at school to provide education on teen dating violence. At the societal level, this lack of support could be considered as an underlying structural violence that hinders the promotion of non-violence norms in scholar settings.

In the focus groups, the participants pointed to the overlap of other causes in ineffective coping, such as lack of experience, lack of trust in the family, and inability to perceive a violent relationship. The situation is exacerbated by the influence of a patriarchal culture that ultimately guides how challenges to personal integrity are handled. The testimonials demonstrate how violence is normalized, violent behavior is justified, victims are blamed, and relationships are idealized in a way that follows the patriarchal model, as demonstrated by expressions that support the myths of romantic love.

The findings coded as “patriarchal culture” are consistent with those of various studies that have found an association between violence in dating relationship, sexism, and romantic love [7,19]; an association between romantic love, blaming the victim and legitimization of the violence [20]; an intergenerational transmission of violence, where it is learned that violence is appropriate for conflict resolution [21,22,23]; and the belief that “good jealousy” exists, understood as a game that serves to measure how much love or interest one person, when provoked, has for the other, who has initiated the game, which is common among adults [24]. We learned in the focus groups that this pattern is being reproduced by adolescents.

The combinations of the analyzed factors lead to adolescents having difficulty responding to dating violence. This result is in line with a previous study that demonstrated positive correlations between romantic myths and cyber dating violence victimization [25]. The reality contrasts with isolated experiences, where the focus group participants mentioned talking more openly about dating with their parents or significant others; this support is fundamental, and the participants suggested using a guide to educate parents and guardians on this issue.

Another important finding from the focus groups is the need to receive training to promote personal and familial strategies to prevent violence. The most frequently related codes corresponded to family communication, examples at home, and dialogue spaces, which allow adolescents to talk and share experiences [26]. The adolescents were interested in receiving advice that would allow them to reflect and make decisions if conflict occurred during dating. The promotion of egalitarian attitudes in the couple has been related to the recognition of abuse, which is essential for its approach [24]. During the focus groups, behaviors that reinforced health-promoting behaviors were identified by the adolescents, such as respect, trust, balance, valuing oneself, and setting boundaries in the relationship.

In this regard, a recent research has supported the need to offer younger couples adequate tools to solve conflicts in their relationships as a way to avoid reinforcing violent behavior [6].

Regarding recommendations for developing training, the participants emphasized the use of methodologies they are interested in, such as dramatization, which favors the participative creation of positive strategies for conflict resolution [26,27], or incorporating information and communication technology in prevention programs that promote healthy and non-sexist relationships. This result reinforces the notion that effective education and gender education [28] contribute to reductions of abusive practices.

Undoubtedly, these findings will have implications for nursing practice. Those are related to the expansion of nursing actions and service demands, such as developing interventions focused on roles and relationship patterns that support the recognition of emotional responses or reactions when facing different situations in daily life. Adolescents’ exposure to violence, either as victims or perpetrators, necessitates the teaching of constructive coping skills for conflict resolution and modifying the perception of the problem, as it is usually linked to cognitive distortions or false beliefs about romantic love that follow traditional gender roles. It is also important to provide guidance for adolescents on how to find support systems, available resources, family coping strategies, peer groups, and school environments to prevent dating violence.

One limitation of this study is that the research method does not allow the findings to be generalized; however, the results are in line with the literature found, and they enable the reality experienced by adolescents and the development of prevention programs to be explored. The focus group technique does not allow delving into personal experiences, although it offers benefits in the evaluation of collective subjectivity. The mixed-gender focus group was productive in offering a mutual characterization of both important perspectives (male and female). Even though it is possible that it affected both the content of the discussion and the dynamics of interaction in the group, the perception of the moderator and the observer is that the participants felt comfortable and that participants of both sexes made similar use of time during the discussion.

## 5. Conclusions

The assessment of roles and relationships, as well as coping mechanisms, is a valid way to approach adolescents and learn how they perceive interaction during dating and which strategies are used most frequently to prevent violence. Assessment is the starting point for designing interventions that promote healthier personal relationships, and both phases (assessment and intervention) can be carried out by interdisciplinary teams or those comprising nursing professionals.

The data collection method used in this study facilitated adolescents’ participation in both the evaluation processes and in the creation of proposals that could be incorporated into the intervention plan.

Dating violence is a problem that requires the design of educational proposals that are aligned with adolescents’ concerns and interests. These should utilize dialogue spaces and interactive tools that promote self-awareness and provide reliable information that highlight gender perspectives, the characteristics of healthy relationships, life and communication skills, types of dating violence, and the importance of seeking help. This last aspect is crucial for the promotion of self-care practices and taking responsibility for one’s health. Additionally, it is a necessity to expand nursing interventions to the family to achieve the support emphasized and recommended by the participants. Future research can focus on developing and evaluating such dating violence prevention programs.

## Figures and Tables

**Table 1 ijerph-20-02355-t001:** Predetermined codes: Deductive stage.

Predetermined Categories	Definition	Approximating Anchors
Patriarchal culture	Cultural norms that explained the occurrence of the intimate partner violence.	Normalization of violence, silence, myths of romantic love, and justification.
Coping with dating violence	Behavioral tactics used to manage violence in teen dating.	Seeking support to decrease emotional distress
Strategies for preventing dating violence	Personal, familial, and collective actions recommended to prevent teen dating violence	Communication, education, and campaigns to prevent teen dating violence.

**Table 2 ijerph-20-02355-t002:** Participant Characteristics.

Educational Institution	Sex		School Year
	Girls	Boys	
K-12 School (D1)	6	6	Third year of high school
Private K-12 School (D2)	8	6	Second year of high school
Institute (D3)	3	3	Ninth course of basic general education
High School (D4)	5	4	First year of high school
Public K-12 School (D5)	6	3	Second year of high school
K-12 School (D6)	6	1	Ninth course of basic general education
Community K-12 School (D7)	6	5	First year of high school
K-12 School (D8)	5	5	Third year of high school

**Table 3 ijerph-20-02355-t003:** Factors that hinder the prevention of and coping with violence.

Categories	Subcategories	Testimonial	Participant
Fear		…Those who see me think that since I practice karate, I can defend myself from everything…. In my relationship, I had to put a stop to it because otherwise, I could be mistreated…some people don’t act right away because they don’t know how to do it, they don’t know how to handle it, and others don’t act out of fear…. There are things that are very scary.	PF12-D1
Threats		They say “I’m going to die if you leave me, if you leave me, I’ll kill myself,” and the other person feels compassion for them and stays.	P2M-D3
		It escalates so much that there are times when they use threats…saying things like, for example, “you’re never leaving my side” and “if you leave, I’ll hurt you and your whole family.”	P7M-D1
Lack of knowledge and resources to take action		It’s happened to me…and when you see that the person is crossing a line. The truth is that in that situation…. I don’t know how to act….	P4F-D5
		It makes me ashamed because it was the first time, and I would have liked to have had information to know how to handle it, to know how to react.	PF12-D1
Not identifying being in a violent relationship		I think the first step is to know how to identify when you’re being subjected to violence. It’s important to realize that you’re being attacked so you can stop it…. It’s the first step, it’s the way to stop normalizing that situation.	P13M-D1
		I think it’s hard when someone has a boyfriend/girlfriend for the first time and has a violent experience…. The first love is always something unique, beautiful, it fills you, and because it’s something new, you want to stay in love with someone…. But sometimes, some people don’t realize it and accept violent relationships…they get used to it…they think it’s normal, and they can continue looking for the same kind of people…. They don’t have another perspective because they don’t have any other experiences.	P1F-D1
Shame and concealment of violence		I think many people are embarrassed to talk about these things…but it’s important to stop it, and when it first happens, you have to have the strength to say, no more. You have to understand that if you’re attacked once, that person will always do it. If you don’t stop it in the beginning, you condemn yourself.	P12F-D1
		Because in most cases of violence in relationships, in my opinion, they don’t want anyone to know. Then they don’t go to a place where they can help.	P2M-D5
Immaturity/Lack of Experience		We’re not going to be completely mature like adults…because there’s always something we don’t know, and we can act in the wrong way.	P8M-D5
		It’s also related to experience, because if you’re too young to know what it’s like to have a relationship, I don’t think you should get involved if you’re not ready yet, because you’re still really immature.	P3F-D7
Lack of communicational resources		Let’s say, for example, if you’re a person who gets really angry, fix that.	P7M-D3
Lack of trust and communication with family		Parents have a lot of influence on making good decisions, but in my case, I don’t have a close relationship with my dad or my mom. So, in my case, I have to fix things myself, and sometimes I have to trust other people. I don’t see this as a good option, though, because although they’re close to me, they’re not the same as parents. There’s a conflict, within yourself…will the decision be ok?	P9F-D5
		In families right now, there’s like a taboo of being on the phone; I get home, my dad tells me I’m beautiful, but he’s on the phone, and we also arrive and are on the phone, and we get more information from social media than our parents. So, I think our parents should talk to us and give us advice.	P5F-D8
Stay silent		I think there are many people who accept being treated like this in a relationship, they don’t tell anyone, and so they suffer. Not only are they sad, but they can also have psychological damage. They stay quiet and don’t say anything about how they feel.	P1M-D5
Patriarchal culture	Normalization of violence	It’s also important to ignore what people say; for example, suppose that a person is attacked, beaten, and psychologically abused in a marriage, and the relatives tell them: “You have to endure, in my time, we did that, so you also have to take it, you have to understand that the man is superior, stronger, and sometimes reacts badly because he’s impulsive.” You have to overcome that questioning and move forward with the choice not to accept it.	P6M-D1
		I think it’s hard when someone has a boyfriend/girlfriend for the first time and has a violent experience. The first love is always something unique, beautiful, it fills you, and because it’s something new, you want to stay in love with someone…. But sometimes, some people don’t realize it and accept violent relationships…they get used to it…they think it’s normal, and they can continue looking for the same kind of people…. They don’t have another perspective because they don’t have any other experiences.	P1M-D1
	Myths of romantic love	It’s the same when you’ve been with someone for a long time, you feel that need to be with that person more because of what you feel about the relationship, because of the time you have together and the story you have, these are the things that matter.	P7M-D1
		Keep going because if you love each other, it’s for something, it’s because you decided to be together.	P9M-D2
		If you say she belongs to me, she belongs to me, it’s like a little messed up. It depends on the reason, and it depends on the moment.	P2M-D3
	Paternalism and the love–abuse link	They adopt behaviors believing that it’s for the good of the relationship or for the good of the people…people who are manipulated believe it…they believe that the words they say to you or the times they mistreat you don’t matter because you believe that they do everything for your own good.	P8M-D2
		So, even though there’s violence, almost always the relationship continues, the verbal abuse doesn’t matter because love is stronger.	P1M-D3
	Tendency to blame the victim	I also agree with the idea that someone who allows themselves to be mistreated or manipulated is also pretty guilty…because if you want it to stop, you have the right to say, “No, I don’t want to be mistreated” or something like that…you have the obligation to speak up and say “that’s enough.”	P11F-D1
		You keep that mentality…it’s like a wound. At night, you think about “What did I do wrong to be treated like this?” Or crying every night…it’s like feeling guilty that the other person treats you badly.	P4F-D5
	Justification of violence	Sometimes the woman is to blame, too. Like, because, like, in my case, my mom talks to men and things like that. I mean, it’s different with my dad, so…. I mean, my dad was hitting my mom. That confirmed it. […] He didn’t want to stop fighting at all because he told me, “You, go to your room, don’t get involved.” Anyway, I didn’t. I can’t do anything.	P4M-D6
	Repetition and teachings of sexist patterns at home	I think that people are going reflect what happened at home…they’re going to act based on what they see in their family…it depends on how they were treated by their fathers, and that’s why some men are so violent, because they think that the way they were treated by their parents, maybe they are that way, and they don’t know how to be any other way, so they just copy that behavior….	P3M-D5
		When you see your parents start fighting, insulting each other, or if they start hitting each other…. You think that’s normal, and so sometimes people behave like that in a relationship…because you think it’s normal.	P5M-D2

Triangulation between the researchers based on the participants’ experiences as narrated in the focus groups.

**Table 4 ijerph-20-02355-t004:** Help-seeking behaviors.

Categories	Testimonial	Participant
Do not isolate yourself and tell someone about your situation	I told my friends and my family.	P2F-D6
Ask for advice	Going with my partner to talks, like, they advise us on how the relationship should be.	P3F-D6
Turn to someone with experience	Ultimately, one shouldn’t make the decisions. Well, you have to make them. But after talking to someone who’s more experienced and older. And that person tells us, “Look, you know what, do this.”	P4M-D7
Turn to mothers	I personally do tell my mom my stuff…because I know she always has the right words or advice for me.	P4F-D5
More knowledge	To know more about the subject.	P1F-D8
Reporting the situation	Something like that happened, but with a classmate, and we went to the psychologist to tell her what had happened, and from there, the professionals started to ask for more statements.	P6F-D7

Triangulation between the researchers based on the participants’ experiences as narrated in the focus groups.

**Table 5 ijerph-20-02355-t005:** Personal and familial actions recommended to prevent dating violence.

Categories	Testimonial	Participant
Communication with the family	One’s family has to talk to each other, because if we don’t talk, we think what’s happening is normal. Like they said, it could be a game or something. But in reality, it’s not a game. And sometimes, they even force them to shut up.	P3F-D8
	Parents have a lot of influence on making good decisions, but in my case, I don’t have a close relationship with my dad or my mom. So, in my case, I have to fix things myself, and sometimes I have to trust other people. I don’t see this as a good option, though, because although they’re close to me, they’re not the same as parents. There’s a conflict, within yourself…will the decision be ok?	P9F-D5
Reestablishing communication with the family	There needs to be a workshop like this for parents. And, I don’t know, a psychologist who advises on how to talk to your child and vice versa, a psychologist who advises a child on how to talk to their parents so there’s family communication.	P2F-D8
	I think the school should talk to the parents, in like a parents’ meeting, which, I don’t know, like for them to tell them they need to talk to their kids, because there are some people who, like I said, are only on their phones, on their phones.	P3F-D8
Talk to people with more experience/professionals	If I have a friend who’s older than me, just because she’s older than me, I mean, she’s going to know more than me, so I’m going to follow in her footsteps. I see that she’s doing well doing what she’s doing, and how she’s older than me. Maybe when I’m her age, I’ll do the same.	P2F-D8
Create spaces for dialogue	I think that more spaces like this are really good…because here, you can look for solutions, you can look for help and say what’s happening, when you say what you feel, you realize you’re being harmed. The moment that you can let out what you feel, you realize what’s right and what’s wrong.	P3F-D1
	I wanted to come because I like to hear what everyone thinks and express what I think…that’s why I said it felt like being at the UN. I like this type of discussion; in a group, I learn, I can share with my classmates and friends.	P1M-D5
	I think it’s really good to talk about what’s really happening because you can understand it better, we understand, and we can fix the behaviors that can become violent.	P3M-D5
Example at home	I think that what somebody sees in their family is how they act in real life, that is, when they grow up. And that’s because you believe that or you think that; for me, my dad hit my mom, and my dad is doing well. So, I’m going to hit my girlfriend so that I can do well. I mean, because once I hit a woman, the woman is going to feel, like, so, so harassed that she is not going to separate from that person…. It scares her.	P4M-D8
Education at home	A friend of my mom’s, her husband drank a lot, and one day, the husband got upset and left her face all bruised, all swollen. The poor thing was hardly recognizable. But I only saw photos, I didn’t see, but I heard more or less, and my mom told me that to avoid being like her, it’s better not to be with men like that.	P2F-D7
	My mom gives me a lot of advice because she’s older and has been through a ton of things. She advises me every day. Look, take me as an example. She tells me, “You want to be just like me? Do you want to end up the way I am?” And she tells me so many things.	P3F-D8
Training	With these kinds of discussions, you get my attention…especially the dynamics, that’s what’s important.	P5F-D5
	Include the parents, and our families and siblings, too, because the same things happen to them.	P2M-D6
Coping with social pressure/eliminating the patriarchal system	It’s also important to ignore what people say; for example, suppose that a person is attacked, beaten, and psychologically abused in a marriage, and the relatives tell them: “You have to take it, you have to understand that the man is superior, stronger, and sometimes reacts badly because he’s impulsive.” You have to overcome that questioning and move forward with the choice not to accept it.	P6M-D1
	I think that people who get carried away by stereotypes are violent…men who, in order to be accepted, are sexist, are used to mistreating or seeing mistreatment in their families….	P8F-D2

Triangulation between researchers based on participants experiences narrated in focus groups.

**Table 6 ijerph-20-02355-t006:** Methodologies recommended for education in this issue.

Categories	Testimonial	Participant
Use of social networks	I would give information through social networks. I think it’s what’s used the most, young people are aware of that, through YouTube, some interesting program…. I don’t know. But with information about dating conflict using social networks.	P2M-D8
Visual material	People talk a lot about confidence and communication…but it would need to be more experiential, less theoretical…maybe expose cases of situations that happen that we’ve seen and that are familiar to us, because young people respond more to visuals than talks…because they listen but don’t understand them…so, they should capture our attention, be eye-catching, so we can put it into practice in our lives.	P5F-D5
Talk about experiences that help identify violent relationships	Many professionals talk as if they had memorized a list of things, and that’s boring, I’d like it to be more natural, that they talk about experiences.	P1M-D5
	I think the first step is to know how to identify when you’re being subjected to violence. It’s important to realize that you’re being attacked so you can stop it…. It’s the first step, it’s the way to stop normalizing that situation.	P13F-D1
Dramatization or theater	Maybe a dramatization, like a visual representation of what can cause, for example, a small push at the beginning of the relationship to end in the death of the relationship, in serious cases, if it isn’t stopped in the beginning. Something like that.	P5F-D7
Working on healthy relationships while dating	I think that respect for the other person, for their decisions and tastes, so that both people feel good.	P5M-D5
	The main thing in a relationship is trust and above all honesty…. Those are expressions of love instead of being like: “Give me your password.”	P4F-D2
	My mom tells me that you have to have a balance of everything. If he gives you something, I give it, too. If he gives me something, I give it, too. And if he doesn’t give it, I don’t give it, either. Because my mom says that if he treats me badly, I can’t treat him like the best thing in the world, either.	P3F-D8
	It’s important to value ourselves, we have to know and trust how much we are worth, how much I should love myself, and how much I deserve…we should respect ourselves because if you don’t respect yourself, no one will.	P1F-D1
	Or also, like…for example, returning to the topic of signs of love (the boyfriend asking to have sex as a sign that the girl loves him). For me to say no, not yet, that’s a boundary that has to be made, my respect.	P3M-D8
Campaigns	Preventive campaigns on what can cause dating violence. Make campaigns, let’s tell a couple that they can count on help from the police or people, let’s tell them in cases of physical abuse, yes, you can count on help and that people will help you.	P6F-D7

Triangulation between the researchers based on the participants’ experiences as narrated in the focus groups.

## Data Availability

Data is unavailable due to privacy or ethical restrictions.

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
