# Peer review of "Strategies to Prevent and Cope with Adolescent Dating Violence: A Qualitative Study"

_ijerph, 2023, doi:10.3390/ijerph20032355_

Round 1

Reviewer 1 Report

I congratulate the authors for this qualitative research that generated useful results in practice for the creation of strategies to prevent and cope with dating violence among adolescents.

However, in my opinion, the article requires some revisions/clarifications before being accepted for publication- please see below.

- The authors must clarify the age of the study participants. In the abstract and in the Results section it is mentioned that the participants were aged between 13 and 17 years, and in the Materials and Methods section it is mentioned that the participants were between 13 and 19 years old.

- The number of participants in each focus group must be revised- in the abstract, the authors mention that “Eight focus group discussions were analyzed, which included a total of 78 adolescents […]" (Lines 14-15), while in the Materials and Methods section, the authors mention “Data were collected through mixed-gender focus groups, which included a minimum of six participants and a maximum of nine” (Lines 115-116). Mathematically, even if the number of participants in each focus group had been maximum, i.e., 9 participants, it would not have been possible for a total of 78 teenagers to participate in the study.

- Line 59- “Regarding strategies to prevent dating violence. A systematic review of 10 studies of […]”- the full stop between the two sentences should be deleted

- The tables within the Results section should be revised as in some places the same quotation is attributed to two different participants

For example: 

·      Table 3- Category “Not identifying being in a violent relationship”- quotation attributed to participant P1F-D1; Category “Patriarchal culture”, sub-category “Normalization of violence”, the same quotation is attributed to participant P1M-D2. 

·      Table 3- Category “Patriarchal culture”, sub-category “Normalization of violence”, the first quotation is attributed to participant P6M-D1; in Table 5- Category “Coping with social pressure/eliminating the patriarchal system”, the same quotation is attributed to participant P6M-D2

- Sub-sections 3.2.2 and 3.2.3 have the same title. Please correct!

- Paragraphs L226-238 and L253-265 are identical. Please delete one of them.

- Paragraph L 282-285 is not clear. Please revise it!

- Paragraphs L 305-306 and L 319-322 refer to the results of the study. Not clear why these paragraphs include bibliographic citations

- Lines 323-324, 341, 351- The authors specifically mention the role of nursing professionals in preventing and coping with dating violence in adolescents, although their role does not result from the study. Please provide clarification on this aspect. 

Author Response

Response to Reviewer 1 Comments

Point 1: The authors must clarify the age of the study participants. In the abstract and in the Results section it is mentioned that the participants were aged between 13 and 17 years, and in the Materials and Methods section it is mentioned that the participants were between 13 and 19 years old.

Response 1: Thank you very much. It was modified. (in red) Line 91 and 99.

Point 2: - The number of participants in each focus group must be revised- in the abstract, the authors mention that “Eight focus group discussions were analyzed, which included a total of 78 adolescents […]" (Lines 14-15), while in the Materials and Methods section, the authors mention “Data were collected through mixed-gender focus groups, which included a minimum of six participants and a maximum of nine” (Lines 115-116). Mathematically, even if the number of participants in each focus group had been maximum, i.e., 9 participants, it would not have been possible for a total of 78 teenagers to participate in the study.

Response 2: Thank you very much. It was modified. (in red) Line 114.

Point 3: - - Line 59- “Regarding strategies to prevent dating violence. A systematic review of 10 studies of […]”- the full stop between the two sentences should be deleted.

Response 3: Thank you very much. It was modified.

Point 4: - - The tables within the Results section should be revised as in some places the same quotation is attributed to two different participants. For example:

Table 3- Category “Not identifying being in a violent relationship”- quotation attributed to participant P1F-D1; Category “Patriarchal culture”, sub-category “Normalization of violence”, the same quotation is attributed to participant P1M-D2.

Table 3- Category “Patriarchal culture”, sub-category “Normalization of violence”, the first quotation is attributed to participant P6M-D1; in Table 5- Category “Coping with social pressure/eliminating the patriarchal system”, the same quotation is attributed to participant P6M-D2

Response 4: Thank you very much. It was modified. (in red) Table 3

Point 5: -Sub-sections 3.2.2 and 3.2.3 have the same title. Please correct!

- Paragraphs L226-238 and L253-265 are identical. Please delete one of them.

Response 5: Thank you very much. We deleted paragraphs Lines 253-265.

Point 6: a - Paragraph L 282-285 is not clear. Please revise it!

b-Paragraphs L 305-306 and L 319-322 refer to the results of the study. Not clear why these paragraphs include bibliographic citations

Response 6: Thank you very much. We included a short explanation about the asociation with the study before published.

  1. a. We rewrote paragraph (in red) Lines 276-281.
  2. b. We rewrote paragraph (in red) Lines 323-327.

Point 7: -Lines 323-324, 341, 351- The authors specifically mention the role of nursing professionals in preventing and coping with dating violence in adolescents, although their role does not result from the study. Please provide clarification on this aspect.

Response 7: a. We rewrote paragraph (in red) Lines 329-330. b. We have considered that is not part of the study but it is a orientation for future interventions in the nursing practice.

Reviewer 2 Report

this article addresses the topic of violence and personal injury. violence in dating is a historical theme that finds wide scientific resonance. This article is well written, data are well represented

Author Response

Response 1: Thank you very much for your comments.

Reviewer 3 Report

The authors have conducted a valuable piece of research which aims to have an impact on the design of future prevention and intervention efforts that aim to promote adolescents’ constructive use of coping mechanisms in response to dating violence. This was a well-designed study with a healthy sample size and was very interesting to read, a welcome piece in an area that has been recognised as a public health concern and in need of further work on this issue and how to address it in society more effectively. I feel that this is suitable for publication after some revisions.

Page 2 lines 49-51

·       Studies have found that girls are more likely to be the victims of serious violence from boys?

Page 2 line 53

·       Spelling error ‘expose’ should be ‘exposed’?

Page 2 lined 54-55

·       Spelling errors ‘romanitcs relationship’ should be ‘romantic relationships’

Page 2 line 63

·       Mistrust mentioned twice in same sentence

Page 2 fourth para lines 67-71

·       These sentences don’t seem to flow. First sentence talks about coping strategies and then the next talks about causes of mental health problems?

Page 2 line 72

·       Comma not needed after 'relevance'

Page 3 line 128

·       “All the interviews were audio recorded by the moderator and research group” – who is the ‘research group’? Do you mean the moderator and the observer?

Methods

·       More detail about the methodological procedures and processes would be good. E.g. the creation of the categorical framework and pre-determined codes etc.

RESULTS – page 4+

·       While very interesting and I like the use of quotes to support the themes, I found the way this was presented little confusing and wonder if this could be organised more clearly and coherently in a logical structure…  e.g. it starts off talking about the immature theme in the narrative/text but then the other themes are just presented in a table. Consistency is needed I think – either just use the table and quotes or discuss each theme and have the table just to show the theme/subtheme titles  – it just felt like it could flow / be organised better….

Results - page 8

·       Again, the discussion of the results felt like it could be structured more coherently. The discussion of the themes in the table is quite brief. And there are points highlighted in the text that are not listed in the table e.g. the discussion of strategies that ‘constituted changing their own behaviour in the relationship to please their partner and thus encourage them to abandon violent attitudes’ (line 228-229).

Results – page 9 and page 10

·       Tables 5 and 6 both have the same title. They need to be distinct. Why are they separated if they are on the same topic? is the title of the latter one incorrect?

Page 8 lines 226-238 and page 10 253-265

·       The text and quotes are repeated?

Page 11 lines 282-284

·       I thought this sentence was a little unclear and could be re-written more clearly.

Discussion section

·       More reference to other studies with similar/different findings would be good. This is done well for the patriarchal culture theme but not so much the other findings.

·       Could consider the role of dramatization/theatre as a preventative intervention in more depth in the discussion – I think this is an interesting development and has been shown to be quite effective in other contexts…

·       Some of the subthemes within the patriarchal culture category are very concerning, yet not entirely surprising, and highlight some serious implications for adolescents’ understandings of healthy relationships, dating violence, and victim blaming attitudes. I would have liked to have seen more consideration for this in the discussion

·       Consideration for the implications of conducting mixed-gendered focus groups would have been nice to include. Not saying that isn’t appropriate or useful, in fact both have strengths and limitations in doing mixed and same-gendered groups, but reflections on the methodology would be nice. e.g. would the findings have been different / discussion if they were same-gendered groups?

The limitations section of the discussion is very brief, perhaps this could be considered in more depth.

·       More detailed consideration and suggestions for further research would be nice.

Author Response

Response to Reviewer 3 Comments

Point 1: Page 2 lines 49-51 - Studies have found that girls are more likely to be the victims of serious violence from boys?.

Response 1: Thank you very much for your observations, we have changed the mistake (in red) Lines 50-51.

Point 2: - Page 2 line 53 - Spelling error ‘expose’ should be ‘exposed’?.

Response 2: Done (in red) Line 53

Point 3: - Page 2 lined 54-55 - Spelling errors ‘romanitcs relationship’ should be ‘romantic relationships’

Response 3: Done (in red) Lines 54-55

Point 4: -Page 2 line 63 - Mistrust mentioned twice in same sentence

Response 4: We erased the repetition (in red) Line 63

Point 5: -Page 2 fourth para lines 67-71 - These sentences don’t seem to flow. First sentence talks about coping strategies and then the next talks about causes of mental health problems?

Response 5: Thank you we changed the sentence (in red) Lines 68-70.

Point 6: -Page 2 line 72 - Comma not needed after 'relevance'

Response 6: Done. Line 72

Point 7: Page 3 line 128 - “All the interviews were audio recorded by the moderator and research group” – who is the ‘research group’? Do you mean the moderator and the observer?.

Response 7: Done. Line 126

Point 8: Methods · More detail about the methodological procedures and processes would be good. E.g. the creation of the categorical framework and pre-determined codes etc.

Response 8: Done (in red) Line 131-133

Point 9: Results - page 8 - Again, the discussion of the results felt like it could be structured more coherently. The discussion of the themes in the table is quite brief. And there are points highlighted in the text that are not listed in the table e.g. the discussion of strategies that ‘constituted changing their own behaviour in the relationship to please their partner and thus encourage them to abandon violent attitudes’ (line 228-229).

Response 9: Thank you very much for the observation. We modified some parts of the results (in red) Line 170-273

Point 10: Results – page 9 and page 10

  • Tables 5 and 6 both have the same title. They need to be distinct. Why are they separated if they are on the same topic? is the title of the latter one incorrect?

Response 10: Thank you very much for the observation. We modified title of both tables.

Point 11: Page 8 lines 226-238 and page 10 253-265

  • The text and quotes are repeated?

Response 11: Thank you very much for the observation. We modified some parts of the results (in red) Line 170-273

Point 12: Page 11 lines 282-284

  • I thought this sentence was a little unclear and could be re-written more clearly.

Response 12: This sentence has been reformulated (lines 278-281) and the paragraph has been expanded for better understanding.

Point 13: Discussion section

  • More reference to other studies with similar/different findings would be good. This is done well for the patriarchal culture theme but not so much the other findings

Response 13: More references to other studies have been added (in red) to improve the discussion section.

Point 14:    Could consider the role of dramatization/theatre as a preventative intervention in more depth in the discussion – I think this is an interesting development and has been shown to be quite effective in other contexts…

Response 14: The role and benefits of dramatization as a preventive intervention has been incorporated and referenced. See lines 324-325.

Point 15:    Some of the subthemes within the patriarchal culture category are very concerning, yet not entirely surprising, and highlight some serious implications for adolescents’ understandings of healthy relationships, dating violence, and victim blaming attitudes. I would have liked to have seen more consideration for this in the discussion.

Response 15: The references and theorizing on these aspects have been expanded in the paragraph in which they are discussed. Line 297-298.

Point 16:    Consideration for the implications of conducting mixed-gendered focus groups would have been nice to include. Not saying that isn’t appropriate or useful, in fact both have strengths and limitations in doing mixed and same-gendered groups, but reflections on the methodology would be nice. e.g. would the findings have been different / discussion if they were same-gendered groups?.

Response 16: we have introduce this reflexion on the limitation section.

Point 17:    The limitations section of the discussion is very brief, perhaps this could be considered in more depth.

Response 17: We have deepened the limitations section. Line 339 to 348.

Point 18: More detailed consideration and suggestions for further research would be nice.

Response 18: More detailed suggestions for further research have been added on the conclussions section.

Round 2

Reviewer 3 Report

A much improved article, many thanks.